# Human herpesvirus 7 integrates into host telomeres via its telomeric repeat arrays

Yingnan Cheng[1,2], Joilson Xavier[1,2], Dusan Kunec[1], Jana Reich[1,2], Christiana Victoria Cismaru[1,2], Dilan Gün Serdar[1,2], Thomas Höfler[1¤], Benedikt B. Kaufer [1,2]*

1 Institut für Virologie, Freie Universität Berlin, Berlin, Germany, 2 Veterinary Centre for Resistance Research (TZR), Freie Universität Berlin, Berlin, Germany

¤ Current address: Department of Diagnostic Medicine and Pathobiology, Kansas State University, Manhattan, Kansas State, United States of America

* b.kaufer@fu-berlin.de

## Abstract

Human herpesvirus 7 (HHV-7) is a ubiquitous betaherpesvirus and a causative agent of roseola infantum. HHV-7 harbors telomeric repeat arrays (TMR) identical to human telomeres at the ends of its genome. While similar repeats contribute to human herpesvirus 6 (HHV-6) integration into host telomeres, HHV-7 integration and the role of the TMR remained elusive. Using fluorescence *in situ* hybridization and nanopore sequencing, we demonstrate that HHV-7 efficiently integrates into host telomeres in persistently infected cells. To determine the role of the TMR in the virus life cycle, we generated the first HHV-7 reverse genetic system and mutants lacking the TMR. These mutants revealed that the TMR are dispensable for HHV-7 replication, but play a crucial role in the integration process and genome maintenance in persistently infected cells. This study provides a reverse genetic system for HHV-7, and offers important insights into the biology of this ubiquitous human pathogen.

### Author summary

HHV-7 infects almost all humans early in life, is a causative agent of roseola infantum and is associated with several other diseases. This study uncovered that HHV-7 efficiently integrates its genomes into the telomeres of host chromosomes in latently infected cells. To investigate the integration mechanism, we generated the first reverse genetic system for the virus. Deletion of the telomeric repeat arrays at the ends of the virus genome revealed that these sequences are crucial for virus integration. This study provides the first HHV-7 genetic system and important insights into the mechanism that allow this ubiquitous human pathogen to persist in the host for life.

**Data availability statement:** The HHV-7-GFP BAC and TMR mutant viruses are available upon request (via email to virologie@vetmed.fu-berlin.de) and were sequenced by nanopore sequencing. The GenBank files are available on NCBI GenBank. The HHV-7-GFP BAC sequence can be found under the accession number PQ722539 at https://www.ncbi.nlm.nih.gov/nuccore/PQ722539.1/. The HHV-7-GFP BAC ΔimpTMR, ΔpTMR and ΔTMR sequences can be found under the accession numbers PV694271, PV694272 and PV694273 respectively.

**Funding:** This work was supported by the European Research Council (ERC) Consolidator grant ENDo-HERPES (ERC-CoG 101087480 to BBK) and China Scholarship Council (CSC No. 202008330353 to YC). The funders had no role in study design, data collection and analysis, decision to publish, or preparation of the manuscript.

**Competing interests:** The authors have declared that no competing interests exist.

## Introduction

HHV-7 is a ubiquitous human pathogen with a seroprevalence of about 90% in humans [1]. Primary HHV-7 infection usually occurs during the first three years of life [2]. The virus is a causative agent of roseola infantum (exanthem subitum), but less frequent than HHV-6B. HHV-7 infection is also linked to various diseases including febrile epilepsy, seizures, encephalitis, meningitis, and skin rashes [3–6].

Upon primary infection, HHV-7 can establish latency which allows the virus to persist in the host for life - a hallmark of all herpesviruses. Most herpesviruses maintain their dsDNA genome as extra-chromosomal circular episomes during latency [7]. To ensure that the viral genome is maintained in actively dividing latent cells, some herpesviruses, e.g., Epstein–Barr virus (EBV) and Kaposi sarcoma–associated herpesvirus (KSHV) tether their circular episomes to host chromosomes [8,9]. We and others have shown that certain herpesviruses can integrate their genome into the telomeres of latently infected cells [10–15]. These include human herpesvirus 6 (HHV-6) [10,11], the highly oncogenic Marek's disease virus that infects chickens (MDV; GaAHV2) [12,13], gallid herpesvirus 3 (GaAHV3) [15], and herpesvirus of turkey (HVT; MeAHV1) [14]. Chromosomal integration ensures that these viruses are stably maintained in latent cells and efficiently passed on to the daughter cells during cell division. Unfortunately, until now the mechanism used by HHV-7 to maintain its genome during latency is not known.

The HHV-7 genome is about 153 kb in length and consists of a unique region (U) and two direct repeat (DR) regions: DR left ($DR_L$) and DR right ($DR_R$). Intriguingly, both DR regions contain two telomeric repeat arrays (TMR), the perfect TMR (pTMR) and imperfect TMR (impTMR), that are identical to the host telomere sequences (Fig 1A). The pTMR are present at the right end of the DR regions and contain conserved (TTAGGG)n repeats. The impTMR are located at the left end of the DR and its telomeric repeats are interrupted by related hexamers [16–19]. In addition, DR regions contain pac1 and pac2 sequences required for cleavage and packaging of the virus genome and the DR1 to DR6 genes. Interestingly, HHV-6A/B, MDV, HVT and GaAHV3 also harbor similar TMR repeats at the ends of their genomes that facilitate their integration into host telomeres [10–15]. However, the roles of the TMR in the viral life cycle of HHV-7 remain completely unknown.

In this study, we investigated if HHV-7 can integrate into the telomeres during the establishment of latently. We could demonstrate that HHV-7 efficiently integrates to maintain its viral genome over time in latently infected cells. To assess the role of the TMR in the virus life cycle, we generated the first reverse genetics system for HHV-7 using TAR cloning in *Saccharomyces cerevisiae*. This system facilitated generation of pTMR and impTMR mutants that enabled investigation of their roles in virus replication. We found that both TMR are completely dispensable for virus replication but are crucial for integration of the virus genome into host telomeres. Our study provides important insights into HHV-7 biology as well as a genetic system that will help advance understanding of the complex biology of this ubiquitous human pathogen.

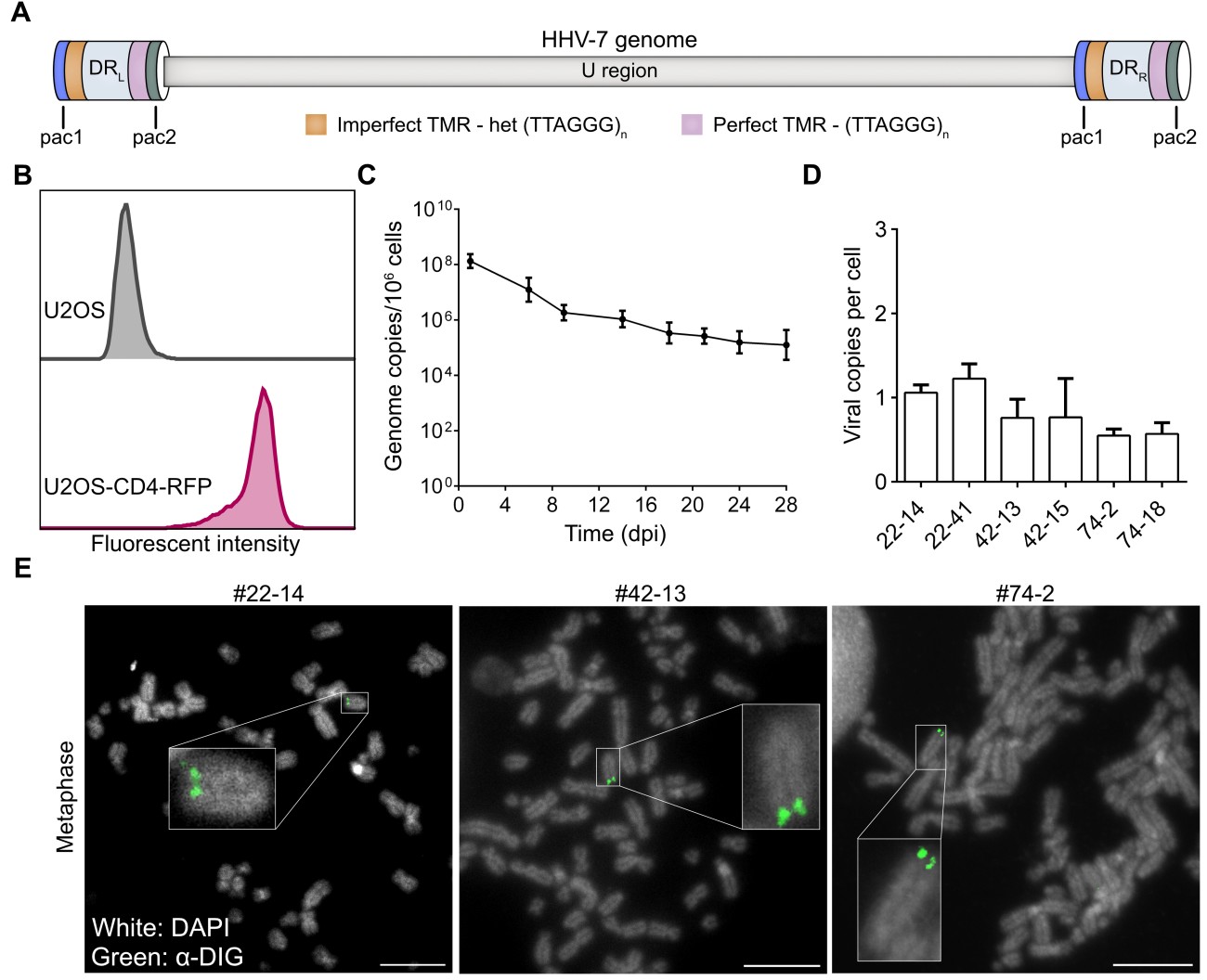

**Fig 1. Integration of the HHV-7 genome in infected U2OS cells. (A)**, Schematic representation of the HHV-7 genome (153 kb) consisting of a unique region (U region) and two direct repeat (DR) regions: DR left ($DR_L$) and DR right ($DR_R$). The DR region contains pac1 and pac2 sequences, the impTMR and pTMR. **(B)**, Flow cytometry analysis of the CD4 expression in the parental U2OS (gray) and the U2OS-CD4-RFP cells (red). **(C)**, Maintenance of HHV-7 genome in infected U2OS-CD4-RFP cells over time. Mean genome copies with standard deviation are shown for the indicated time points. The data represent three independent experiments. **(D)**, Mean virus genome copies in six different clonal U2OS cell lines harboring the integrated HHV-7 genome. U38 probe and β2M probe were used for qPCR assay. **(E)**, Detection of HHV-7 genome integration in persistently infected U2OS cells by FISH. The HHV-7 genome was visualized in indicated clonal cell lines using a DIG-labeled HHV-7-specific probe (green). Nuclei and chromosomes were counterstained using DAPI (white). Almost all cells had clear staining for the virus genome. The scale bar corresponds to 10 µm.

## Results

### HHV-7 can integrate into host telomeres

To assess whether HHV-7 can integrate into host chromosomes, we established an *in vitro* integration assay based on U2OS cells, as previously developed for HHV-6A/B [11,20]. First, we generated U2OS cells constitutively expressing CD4 (U2OS-CD4-RFP, Fig 1B), the cellular receptor of the virus [21–23], to make them susceptible to HHV-7 infection. Upon infection, high virus loads were detected in U2OS cells due to initial viral replication which declined over time. Viral

genome copies stabilized after 18 days post infection (dpi) while no lytic replication was detected, indicative of a persistent infection of the culture (Fig 1C). As less than 10% of the cells harbored the virus genome, we generated cell clones to investigate the state of the virus genome. Clones harboring the virus genome were identified by qPCR, revealing that the clonal cell lines harbored about one copy of the HHV-7 genome (Fig 1D). To determine if HHV-7 integrates into host chromosomes, we performed fluorescence *in situ* hybridization (FISH) of metaphase cells. The FISH demonstrated the presence of HHV-7 genomes in both chromatids at the ends of metaphase chromosomes (Fig 1E) [10,11,24,25]. These data provide clear evidence that HHV-7 integrates into U2OS chromosomes, while the exact location remained to be determined.

### Mapping HHV-7 integration sites by nanopore sequencing

To investigate the HHV-7 integration sites, we performed nanopore sequencing on the persistently infected clonal cell lines. To increase the number of reads covering the target region, we developed a Cas9-based enrichment strategy. Briefly, the purified DNA from the cells was dephosphorylated to prevent nanopore linker ligation. Subsequently, four crR-NAs targeting the U7, U10 and DR6 regions were used to cleave the end of the HHV-7 genome and expose phosphates for the linker ligation and targeted sequencing of the virus genome (Fig 2A). Sequencing of three latently infected clonal cell lines generated long reads (>10 kb) containing the viral $DR_R$ and human subtelomeric sequences (Fig 2B), revealing the orientation of the integrated HHV-7 genome (Fig 2C). Consistently, sequence data showed the viral $DR_L$ is adjacent to telomere sequences measuring at least 8 kb, representing the telomere end of the host chromosome (S1 Data).

Some $DR_R$ reads exceeded 30 kb, allowing the identification of the chromosome harboring the virus genome (e.g., chromosome 6 in case of line 22–14). Between the virus genome and the subtelomeres, a longer stretch of telomeric repeat (~1.5 kb) was present (Fig 2B), indicating that the viral TMR recombined with the host telomeres at this position. Notably, the obtained reads lacked the terminal pac1 (in $DR_L$) and pac2 (in $DR_R$) present at the 5' and 3' ends of the HHV-7 genome (Fig 2B), consistent with a recombination event between the outermost pTMR and impTMR in the virus genome with the host telomeres.

### Generation and reconstitution of the first HHV-7 genetic system

Until now, HHV-7 research has been hampered due to the lack of an HHV-7 reverse genetic system. To investigate the role of the TMR in HHV-7 integration, we established a BAC-based genetic system for the virus by TAR cloning (Fig 3A). Briefly, to facilitate replication of the BAC DNA in both yeast and *Escherichia coli* (*E. coli*), we used a vector containing the yeast centromeric plasmid (YCp) and the BAC replicon (YCp-BAC) [26]. High molecular weight (HMW) HHV-7 DNA (JI strain) was extracted from lytically infected cells. Concatemeric DNA was digested with FspAI to obtain unit length genomes (Fig 3B). As FspAI cleaves the virus genome within the U72 gene, we used a linear YCp-BAC vector with hooks that restore the entire U72 gene during recombination (Fig 3C). This cloning strategy resulted in BAC clones harboring the vector between the U71 and U72 genes. To track virus reconstitution and replication of the recombinant virus, we used a YCp-BAC vector that harbors an enhanced green fluorescent protein (EGFP) reporter gene.

Yeast clones were screened by PCR to confirm the presence of the HHV-7 genome. BAC DNA from positive yeast clones was introduced into *E. coli*, and clones were confirmed by RFLP analyses using various restriction enzymes (Fig 3D). To reconstitute the recombinant virus, the BAC construct (HHV-7-GFP) was nucleofected into SupT1 cells expressing various HHV-7 genes *in trans* and by using an optimized stimulation protocol (Figs 4A and S1). Cytopathic effects (cpe) and green fluorescence were observed upon nucleofection of HHV-7-GFP clones (Fig 4B). Replication kinetics revealed that the HHV-7-GFP replicated comparable to wild-type virus (Fig 4C), highlighting that the insertion of the YCp-BAC vector did not impair virus replication. Our data represent the successful generation of the first HHV-7 BAC reverse genetics system that we used to assess the role of the TMR in virus integration.

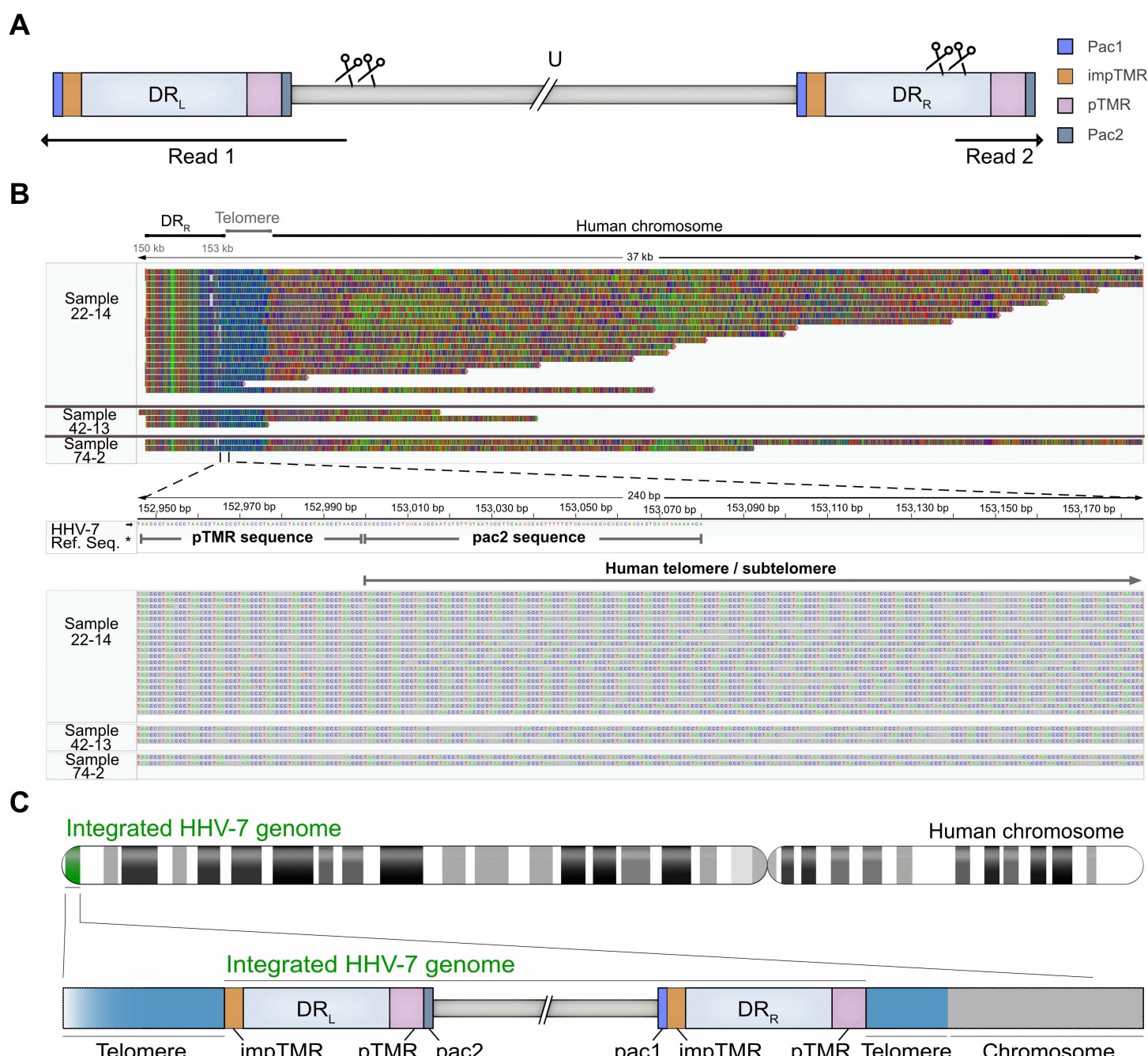

**Fig 2. Nanopore sequencing of the HHV-7 integration sites. (A)**, Cas9-targeted nanopore sequencing of the HHV-7 integration sites. The HHV-7 genome is shown with a focus on the DR regions, U region and the crRNAs (indicated by scissors) used for the cleavage of the virus genome ends. Resulting reads starting in the virus genome are indicated by black arrows. **(B)**, Alignment view of nanopore reads obtained from persistently infected U2OS clones, covering the $DR_R$ end of the virus genome, telomere sequences and the human chromosome with its subtelomere sequences. A zoom into the pTMR/host telomere region is shown, highlighting the absence/loss of the pac2 sequences in the $DR_R$ of the integrated HHV-7 genomes (*HHV-7 reference sequence). Each line corresponds to an independent read. **(C)**, Schematic representation of the integrated HHV-7 genome based on the nanopore sequencing data. The $DR_L$ faces the telomere and the $DR_R$ is oriented towards the centromere. Pac1 in $DR_L$ and pac2 in $DR_R$ are lost during integration, while the internal pac2 in $DR_L$ and pac1 in $DR_R$ remained intact.

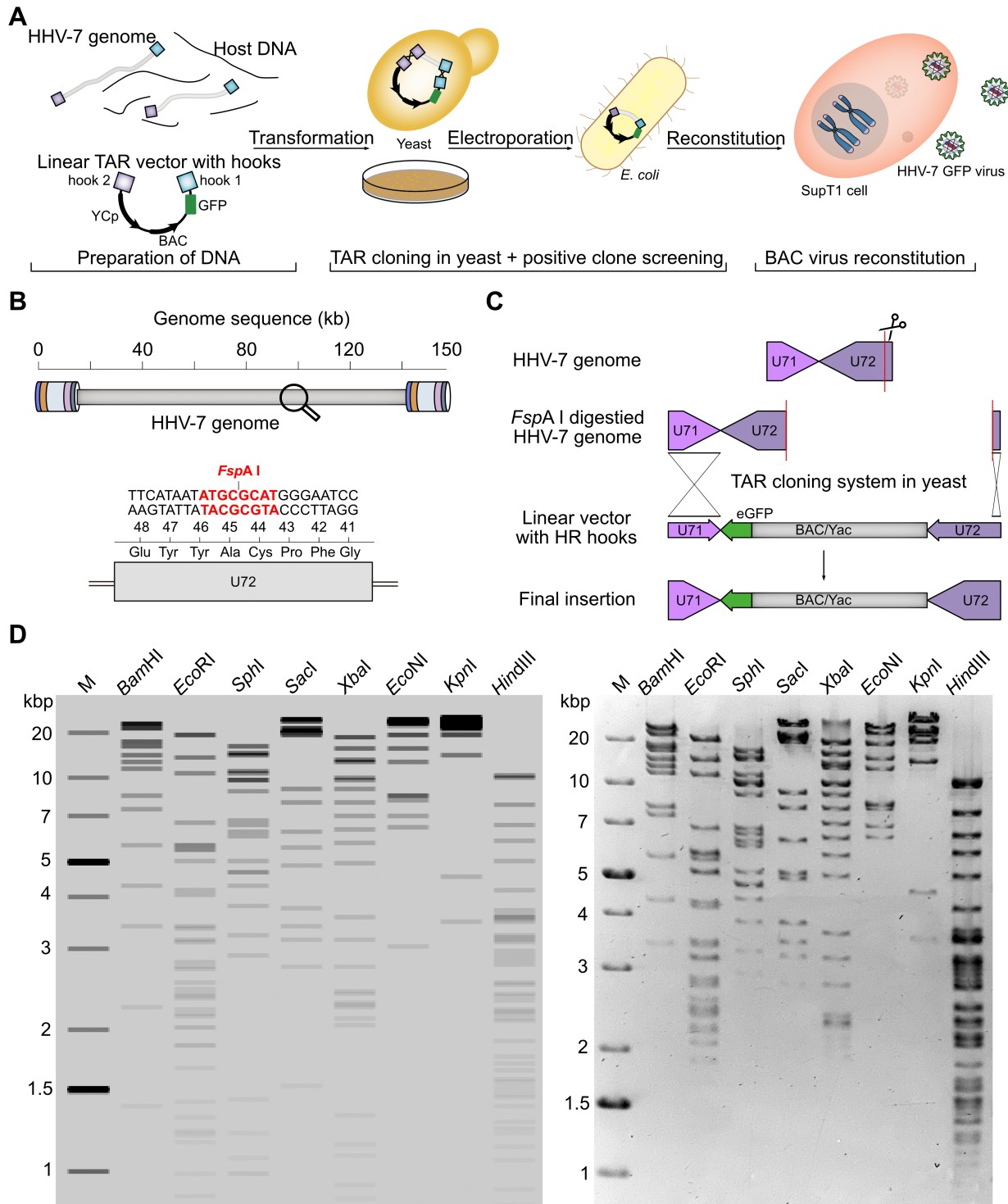

**Fig 3. Generation and validation of infectious HHV-7-GFP BAC clones. (A)**, Establishment of HHV-7-GFP BAC system using TAR cloning in yeast. The linearized HHV-7 genome and the YCp-BAC vector containing sequences homologous to the ends of the HHV-7 genome (hooks) were transformed into *S. cerevisiae* spheroplasts. In yeast, the two fragments recombined to form the circular BAC construct containing the entire HHV-7 genome. DNA from the yeast clones was electroporated into *E. coli* and high-quality DNA was purified. BAC DNA was transfected into SupT1 cells to reconstitute the HHV-7-GFP virus. **(B)**, Concatemeric HHV-7 DNA was cleaved into full length virus genomes using FspAI (red), a unique restriction site that cuts in the

U72 gene. **(C)**, To restore the U72 gene upon FspAI digestion (scissors symbol), the hooks in the YCp-BAC vector contained the entire U71 and U72 regions. This resulted in an HHV-7 BAC with an intact U72 gene and the YCp-BAC vector between U71 and U72. **(D)**, Restriction fragment length polymorphism (RFLP) analysis of the HHV-7 BAC clone using different restriction enzymes. *In silico* prediction (SnapGene) and gel image is shown for the indicated restriction enzymes and the GeneRuler 1 kb Plus DNA-Ladder (Thermo Fisher Scientific).

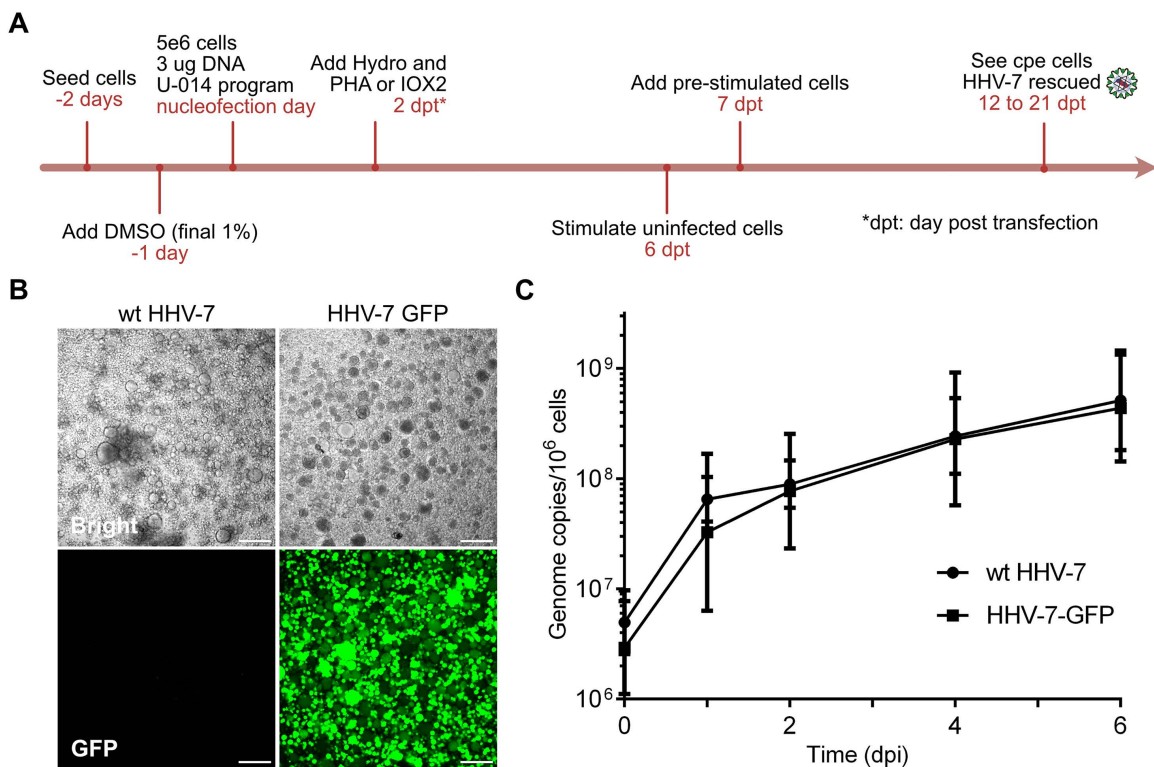

**Fig 4. Reconstitution and characterization of HHV-7-GFP. (A)**, Strategy for the reconstitution of HHV-7-GFP in SupT1 cells using BAC DNA. SupT1 cells were seeded at a density of $5 \times 10^5$ cells/mL two days before nucleofection and DMSO was added to increase the nucleofection efficiency. Nucleofection was performed using $5 \times 10^6$ SupT1 cells and 3 μg BAC DNA per group. Increased GFP expression (S1 Fig) was observed after adding indicated drugs 2 days post-transfection (dpt). Pre-stimulated SupT1 cells were prepared at 6 dpt and added to the transfected cells at 7 dpt. Successful reconstitution of infectious HHV-7-GFP was observed between 12 to 21 dpt. **(B)**, Cpe induced by wt HHV-7 and HHV-7-GFP. GFP expression is clearly detectable in infected cells. **(C)**, Replication kinetics assay of wt HHV-7 and HHV-7-GFP. Mean genome copies with standard deviations are shown for the indicated viruses and time points. There was no statistically significant difference between HHV-7-GFP and its TMR mutants (p-value > 0.05, Wilcoxon matched-pairs signed rank test, n = 3).

## TMR are dispensable for replication but crucial for HHV-7 integration

To determine the role of TMR in the HHV-7 life cycle, we deleted either the impTMR (ΔimpTMR), the pTMR (ΔpTMR), or both TMR arrays (ΔTMR) in the virus genome by *en passant* mutagenesis (Fig 5A) [27,28]. The resulting BAC clones were validated by RFLP (Fig 5B), PCR, Sanger and whole BAC nanopore sequencing (S2 Data). The recombinant viruses were reconstituted in SupT1 cells, replicated efficiently and virus stocks were frozen. Sanger sequencing of DR region of these mutant viruses confirmed the absence of the deleted TMR and the presence of the adjacent pac1 and pac2 sequences as expected (S3 Data). To assess the replication properties of the recombinant viruses, we performed replication kinetics using GFP as a marker of infection. Upon infection, the number of GFP-positive SupT1 cells were quantified by flow cytometry at indicated time points. Replication kinetics revealed that the TMR mutant viruses replicated comparable to the parental virus (Fig 5C), highlighting that the TMR are dispensable for virus replication.

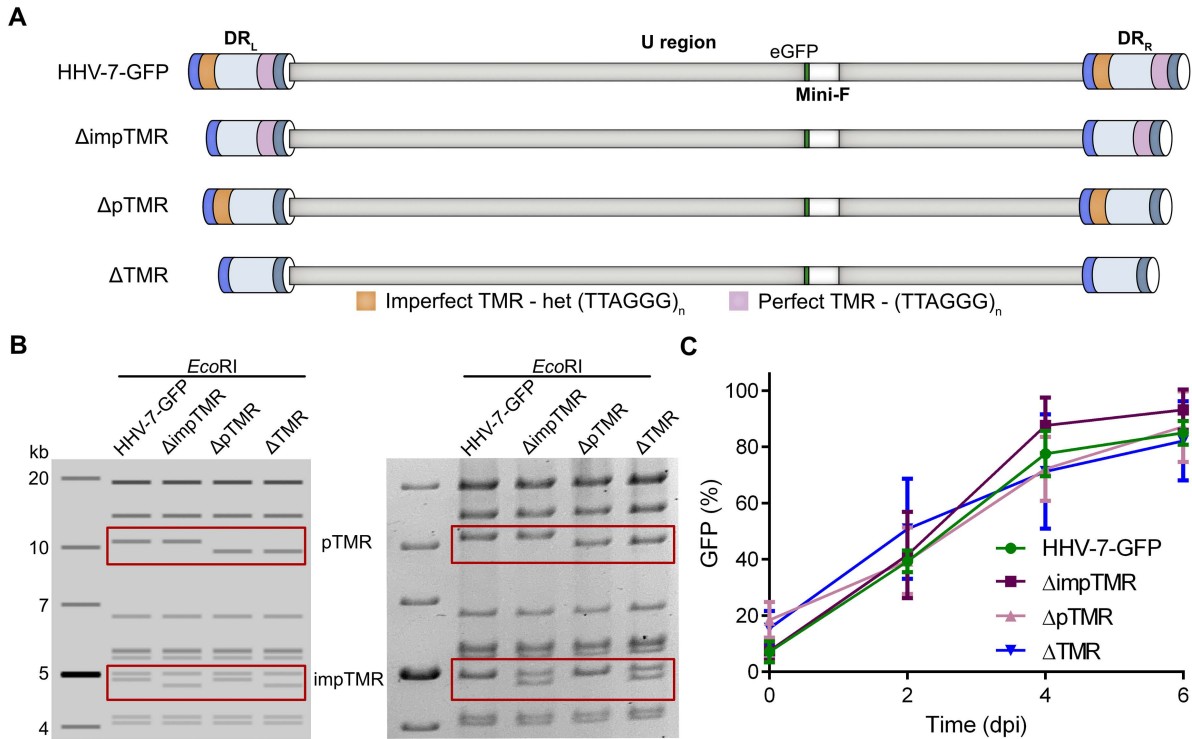

**Fig 5. Generation and characterization of HHV-7-GFP TMR mutants. (A)**, Overview of the recombinant BAC clones with a deletion of either the impTMR (ΔimpTMR), pTMR (ΔpTMR), or both (ΔTMR). **(B)**, RFLP analysis of the indicated clones using EcoRI. Left panel: in silico prediction (Snap-Gene); right panel: agarose gel electrophoresis. The DNA fragments containing pTMR and impTMR are highlighted with red boxes. **(C)**, Replication kinetics assay (cell-to-cell infection). The mean percentage of GFP-positive cells is shown with error bars (standard deviations) for the indicated viruses and time points (p-value > 0.05, n = 3).

To investigate the role of the TMR in HHV-7 genome maintenance and integration, we used our *in vitro* integration assay with one refinement. Upon infection, we sorted GFP expressing U2OS cells (~8% of the population) to obtain a pure population of infected cells and assessed integration and genome maintenance over time (Fig 6A). To determine the role of the TMR in HHV-7 integration, we performed FISH analyses. ΔTMR integration was neither detectable in polyclonal cultures nor in cell clones. However, integrated viral genomes could be detected in both meta- and interphases of ΔimpTMR, ΔpTMR and parental virus (Fig 6B). Integrations appeared to be at the ends of the host chromosomes, indicating that either impTMR or pTMR within the DRs of HHV-7 can facilitate the targeted integration. Next, we assessed the maintenance of the virus genome in the absence of the TMR. While the initial infection levels were comparable between the viruses (Fig 6C), genome maintenance was severely impaired (~150-fold) in the absence of both TMR (ΔTMR) at 28 dpi. The ΔTMR was below to the detection limit in most experiments. A less pronounced reduction was observed upon deletion of only the impTMR (not significant) and pTMR (~2-fold), indicating that pTMR play a role in the integration process and can partially compensate the loss of its counterpart (Fig 6C). In addition, the integration frequencies of the mutant viruses were quantified in metaphase chromosomes by FISH. We observed that the integration of the ΔpTMR and ΔTMR mutant was significantly impaired (Fig 6D). The analysis of interphase nuclei yielded comparable results (Fig 6E). Overall, our data revealed that HHV-7 efficiently integrates into host telomere and that the viral TMR play a crucial role in this process.

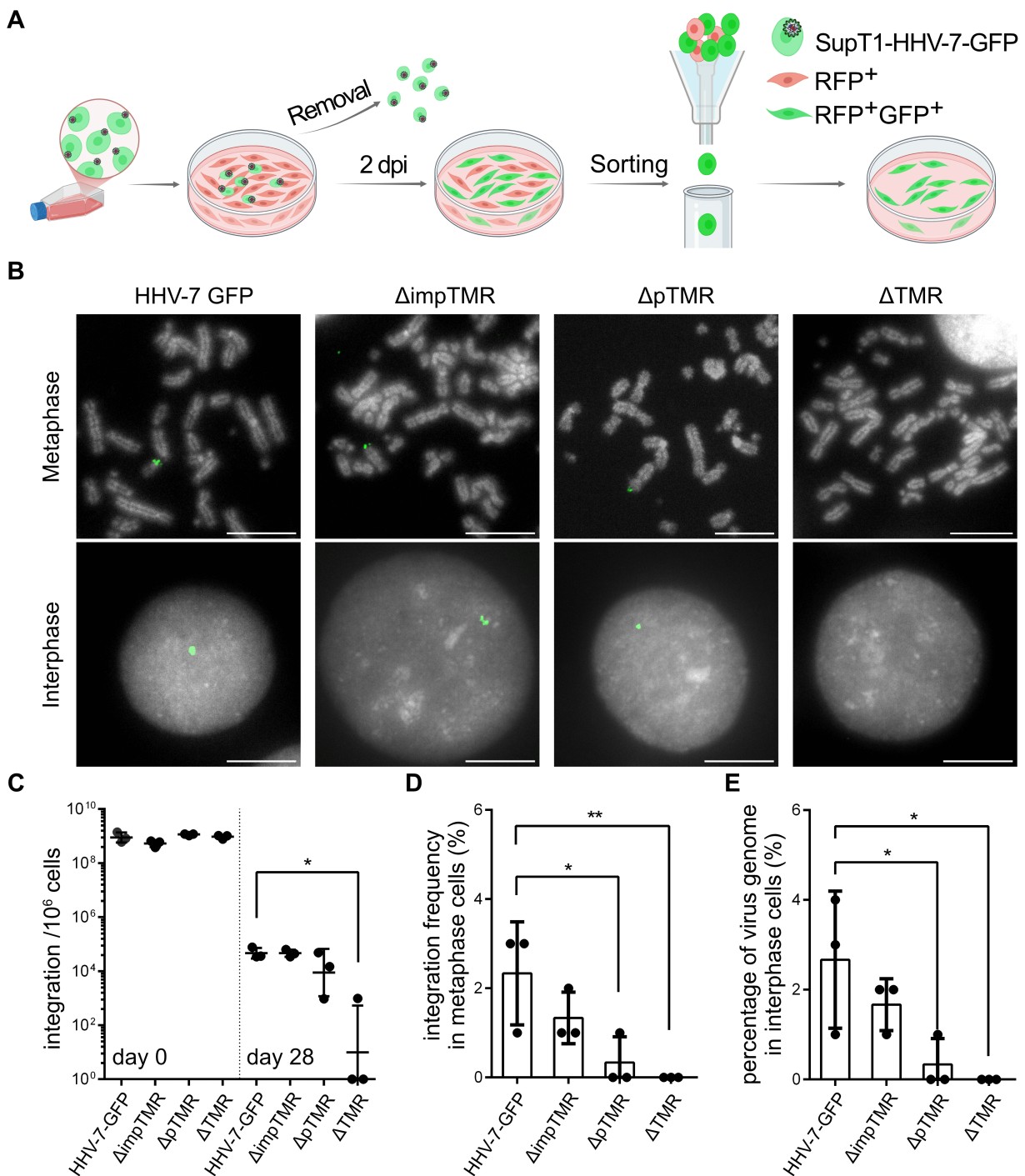

**Fig 6. Integration and genome maintenance in the absence of the HHV-7 TMR. (A)**, Overview of the optimized U2OS integration assay. Assay strategy for monitoring virus maintenance in infected cells. Adherent, confluent U2OS-CD4-RFP cells were infected by seeding $5 \times 10^5$ infected SupT1 cells on top of the monolayer. After 10 hours (h), non-adherent cells were carefully removed. The remaining cells were cultured until 2 dpi, then RFP+GFP+ cells were sorted and subsequently cultured for further analysis. Created in BioRender. You, Y. (2026) https://BioRender.com/iztfzqw. **(B)**, FISH analysis of the HHV-7-GFP and indicated TMR mutant viruses. Representative metaphase and interphase images harboring the virus genome are shown. The Scale bar represents 10 µm. **(C)**, Genome maintenance of the indicated viruses over time. Mean HHV-7 genome copies with standard deviations are shown at day 0 and day 28 after sorting. Asterisks (*) indicate significant differences compared to HHV-7-GFP virus (P < 0.05, Dunnett's

multiple comparisons test, n = 3). **(D)** The integration frequency was determined by analyzing at least 100 metaphases per experiment. Asterisks indicate significant differences between HHV-7-GFP and ΔpTMR, as well as HHV-7-GFP and ΔTMR groups (P < 0.05, Dunnett's multiple comparisons test, n = 3). Data are shown as the means with standard errors. **(E)** 100 interphase nuclei were examined for the presence of HHV-7. Asterisks indicate significant differences between HHV-7-GFP and ΔpTMR, as well as HHV-7-GFP and ΔTMR groups (P < 0.05, Dunnett's multiple comparisons test, n = 3). Data are presented as the means with standard errors.

## Discussion

Until now, it remained elusive how HHV-7 maintains its genome in latently infected cells and whether the virus can integrate into host telomeres. To address these aspects of the HHV-7 life cycle, we established an *in vitro* integration assay using the U2OS cell line. This cell line was previously successfully used to assess the integration of the related betaherpesviruses HHV-6A and HHV-6B [11,20]. However, as these cells do not express CD4 (the only known viral entry receptor of HHV-7) [21], we generated U2OS cells expressing the CD4 receptor. Expression was confirmed by flow cytometry, and allowed an efficient infection of these cells with HHV-7 (Fig 1C, e.g., at 1 dpi). To assess HHV-7 integration, we developed a quantitative integration assay. As HHV-7 is a mostly cell-associated virus, we infected the U2OS-CD4 cells by seeding a defined number of infected SupT1 cells ($5 \times 10^5$) onto the monolayer. This facilitated and efficient infection of the U2OS cells. Due to the high levels of lytic replication in the inoculum, very high viral loads ($10^8$ to $10^9$ copies per million cells) were detected upon infection by qPCR analysis. HHV-7 did not spread to neighboring cells or form plaques on the U2OS-CD4 cells, highlighting that HHV-7 does not productively replicate in these cells. Following the initial infection of the U2OS-CD4-RFP cells, HHV-7 genome copies decreased over time and stabilized after 18 dpi, indicating the establishment of a persistent infection. The virus genome was efficiently maintained until 28 dpi (Fig 1C) and beyond, but could not be reactivated from the clonal cells. This allowed us to assess the integration of the HHV-7 genome in these persistently infected cells. To determine if the virus integrates at the ends of the host chromosomes, we detected the virus genome in metaphase chromosomes. This was initially performed on clonal cell lines harboring the virus genome (Fig 1E), as these were needed for subsequent nanopore experiments. The viral copy number in the clonal cell lines detected by qPCR were less than one copy per cell, which is likely due to the less efficient detection of U38 (standard curve slope = -2.997) vs. β2M (-3.257). Integration of HHV-7 at the ends of host chromosomes was also readily observed in polyclonal latently infected cells (Fig 6B), in the absence of episomal genomes. This is consistent with a previous study detecting the HHV-7 genome at the ends of host chromosomes in SupT1 cells [3]. However, this indirect evidence did not clarify whether HHV-7 integrates into host telomeres or how this might occur. Intriguingly, the integration frequency observed for HHV-7 (2–3%) was slightly lower than the frequency of HHV-6A/B in U2OS cells (5–20%) published previously [22]. The lower integration frequency could be caused by several factors, including differences in i) the integration efficiency, ii) the viral genome (e.g., U94 is encoded by HHV-6A/B but not HHV-7), and iii) expression latency gene that could affect cell proliferation. In case of HHV-6A/B, integration can also occur in germ cells, resulting in an endogenous form of HHV-6 (eHHV-6) [29,30]. While eHHV-6 is present in about 1% of the human population carrying the integrated virus genome in every cell in their body [31], integration of HHV-7 into the germ line is scarcely documented [3]. To our knowledge, none of the large-cohort/population studies observed endogenous HHV-7 in their cohorts. This rare incidence could be due to the inability of the virus to infect germ cells or that cells do not survive HHV-7 infection.

To investigate the HHV-7 integration sites in human chromosomes, we performed nanopore sequencing using the three clonal cell lines investigated by FISH prior to that. To enrich for the HHV-7 genome, we developed a Cas9-targeted nanopore sequencing approach. This allowed us to selectively obtain sequencing reads from the virus-host junctions. These data revealed that HHV-7 specifically integrates into host telomeres. This integration resulted in the loss of the terminal pac1 in the $DR_L$ and pac2 in the $DR_R$, indicating that the virus TMR recombined with the host telomeres [32]. The obtained sequencing reads also revealed the orientation of the HHV-7 genome in the human chromosomes (Fig 2C). The HHV-7 $DR_R$ is facing the centromeres of the chromosomes, while the $DR_L$ end only contains host telomere. Both the

orientation and the loss of pac1/pac2 is consistent with previous data on HHV-6 integration [10,33]. We also used the obtained reads to map the integration to specific chromosomes ends. Due to the similarity between subtelomeric regions, very long reads (>20 kb) are needed to identify a specific chromosome. Even though we obtained long reads covering the integration site, there is still immense potential for optimization of the enrichment strategy to apply it to integration events at a low(er) frequency.

Next, we assessed the role of the TMR in the virus life cycle. Until now, research on HHV-7 has been hampered by the lack of a reverse genetic system. The establishment of (BAC-based) genetic systems are particularly challenging for (mostly) cell-associated and slowly replicating herpesviruses using traditional methods [34]. In this study, we applied TAR cloning to obtain the first HHV-7 genetic system, which only took a few weeks. We isolated concatemeric virus DNA from infected cells and directly captured the entire HHV-7 genome via TAR cloning in *S. cerevisiae* [35–37]. TAR cloning is a powerful genetic tool that utilizes the high efficiency of homologous recombination in yeast to capture (large) desired sequences into a YCp-BAC vector. This technique was used in the early phase of the COVID-19 pandemic to generate a reverse genetic system for SARS-CoV-2 in record time [38]. By cloning the HHV-7 genome into a YCp-BAC vector, we were able to combine the strengths of both domains: yeast and *E. coli*. We used the yeast for rapid assembly of the BAC clone and *E. coli* for precise manipulation of the HHV-7 genome by *en passant* mutagenesis. In contrast to yeast, *E. coli* also allowed the isolation of high-quality HHV-7 BAC DNA required for the reconstitution of the virus.

Even though we successfully generated the first HHV-7 BAC, we were initially not able to reconstitute the HHV-7-GFP virus in SupT1 cells using the traditional phytohemagglutinin (PHA)/IL-2 stimulation method [39–42]. To improve the HHV-7 reconstitution, we generated SupT1 cell lines expressing HHV-7 genes (U18, U53, U73, U90) that could drive virus replication (S1 Fig). Of these, particularly overexpression of U18 and U73 drastically enhanced virus reconstitution; however, the preference of the virus to established latency still slowed down or abrogated virus replication. To overcome these challenges, we used our recently published strategies to enhance HHV-6A reconstitution and replication [43]. Stimulation with PHA or IOX2, a hypoxia mimetic, drastically enhanced virus replication and allowed efficient virus reconstitution. Once successfully reconstituted, replication of HHV-7 can be sustained at high levels using hydrocortisone. These optimizations provided the basis for the efficient reconstitutions of the HHV-7 BAC and the mutants generated for this study.

To assess the role of the TMR in the HHV-7 life cycle, we deleted the impTMR (ΔimpTMR), the pTMR (ΔpTMR), or both TMR arrays (ΔTMR) in the virus genome. Whole BAC sequencing was used to sequence the entire virus genomes, and to ensure that only the desired alterations are present in the TMR mutant viruses. All TMR viruses were efficiently reconstituted from BAC DNA. Replication kinetics also revealed that deletion of TMR had no effect on HHV-7 replication, highlighting that these sequences are dispensable for viral replication. This observation is consistent with the data that we obtained for HHV-6A, MDV, HVT and GaAHV3 in recent studies [10–15].

To investigate the role of the TMR in HHV-7 integration, we performed our integration assays with the TMR mutant viruses. In this case, a defined number of infected SupT1 cells were seeded on the U2OS-CD4 monolayer. Sorting of GFP-expressing cells ensured a pure population of infected U2OS-CD4-RFP cells as the starting material (Fig 6A). FISH analyses revealed that no integration was detected in the absence of all TMR (Fig 6B). Consistently, genome maintenance of ΔTMR was barely detectable at 28 dpi. In most experiments, no HHV-7 genomes were detected, while the residual copies in the other experiments could be explained by an inefficient random integration that that also commonly occurs with plasmid DNA [44,45]. In contrast, integration of ΔimpTMR, ΔpTMR and the parental virus was readily detected by FISH and appeared to be present at the ends of host chromosomes (Fig 6B). The HHV-7 integration frequencies (n = 3) revealed that integration of the ΔpTMR and ΔTMR were significantly impaired, whereas deletion of the impTMR only slightly reduced integration (Fig 6D and 6E). In addition, integration occurred in a wide variety of chromosomes (large, medium and small), suggesting a rather random integration (Figs 1E and 6B). Genome maintenance of ΔimpTMR and ΔpTMR (about 2-fold) was only slightly reduced (Fig 6C), indicating that the loss of the impTMR can be complemented by the pTMR, and vice versa. Intriguingly, deletion and/or mutation of the ΔimpTMR and ΔpTMR counterparts in HHV-6A and

MDV had a more drastic effect on virus integration [10–15]. These findings suggest that the integration mechanism and/or the usage of the TMR differs between HHV-7 and these viruses. Future studies will assess the integration pattern of these TMR mutant viruses by nanopore sequencing, shedding light on the integration mechanism of HHV-7.

In summary, our data clearly demonstrate that HHV-7 integrates into the telomeres of human chromosomes. The TMR in the HHV-7 genome are crucial for this integration process, but are completely dispensable for virus replication. To achieve this, we established the first genetic system for HHV-7 that will help to advance the field and address key knowledge gaps about this ubiquitous virus.

## Materials and methods

### Cells and virus

SupT1 cells were cultured in RPMI 1640 media (Pan-Biotech) supplemented with 10% fetal bovine serum (FBS; Pan-Biotech), 2 mM glutamine (Pan-Biotech), 10 mM HEPES (Pan-Biotech), and 5 µg/mL Plasmocin (InvivoGen). U2OS cells were cultured in a minimum essential medium (MEM; Pan-Biotech) supplemented with 10% FBS. 293T cells were cultured in Dulbecco's modified Eagle's medium (DMEM, Pan-Biotech) supplemented with 10% FBS. All cells were cultured at 37 °C in a humidified atmosphere containing 5% $CO_2$. HHV-7 virus stocks were obtained from the HHV-6 Foundation (https://hhv-6foundation.org/repository) and propagated in SupT1 cells as previously described [40,41,46,47].

### *In vitro* integration assay

To make U2OS cells susceptible to HHV-7, we expressed CD4 (the cellular receptor of the virus) in these cells using a lentiviral expression system as described previously [48]. Briefly, the CD4-RFP cassette from pCD4-TagRFP-T (RHPMP151, addgene plasmid #119238) was cloned into pLenti vector backbone [48] using the NEBuilder assembly kit (NEB). Lentiviruses were produced by co-transfection of adherent 293T cells with pCMVDR8.91, pVSV-G and pLenti- CD4-RFP plasmid at a ratio of 1:1:1, using polyethyleneimine (PEI). The lentiviruses were harvested 72 h post-transfection. U2OS cells were transduced with the CD4-RFP lentivirus using 8 µg/mL polybrene, to enhance transduction efficiency. To further increase the transduction efficiency, lenti-X concentrators (Takara) were used to concentrate the lentivirus stocks according to the manufacturer's instructions.

To assess HHV-7 integration, $2 \times 10^5$ U2OS-CD4-RFP cells were seeded in 24-wells and infected by co-cultivation with highly infected SupT1 suspension cells. $5 \times 10^5$ HHV-7-infected SupT1 cells were seeded on the U2OS-CD4 monolayer and incubated them for 10 h. SupT1 cells were subsequently carefully removed by three stringent washes with PBS. In case of the recombinant viruses, infected GFP+ U2OS cells were sorted using a FACS Aria III cell sorter (BD) at 2 dpi to obtain a pure population of infected cells for the integration assay. Samples were collected at day indicated time points post infection. DNA from virus infected cells was isolated using the Quick-DNA kit (Zymo Research). Viral genome copies were quantified using primers and probes specific for HHV-7 U38 and the cellular β2M gene (Table 1). U38 copies were normalized against the cellular β2M copies.

### Clonal cell lines harboring the integrated virus genome

HHV-7 infected U2OS-CD4-RFP cells were cultured, treated with trypsin, seeded at 0.8 cells per well and cultured for three weeks to obtain clonal lines. Clones were screened by PCR using primers specific for the DR1 gene (Table 1). HHV-7 positive clones were expanded, stocks frozen and analyzed by FISH.

### Fluorescence *in Situ* Hybridization assay

The HHV-7 genome was detected in metaphase cells by fluorescence *in situ* hybridization (FISH), as described previously [11,12,49]. Briefly, the metaphase chromosomes were prepared from HHV-7 positive U2OS clones. HHV-7 probes were

**Table 1. Primers and probes for qPCR-based detection of virus genomes, generation of recombinant viruses and the crRNA for nanopore sequencing are shown.**

| Construct name | | Sequence (5' → 3') |
| --- | --- | --- |
| β2M | For | CCAGCAGAGAATGGAAAGTCAA |
| | Rev | TCTCCATTCTTCAGTAAGTCAACTTCA |
| | Probe | FAM-ATGTGTCTGGGTTTCATCCATCCGACA-TAM |
| U38 | For | TCACGCCCAAGACATGTGA |
| | Rev | GGGCAACACATAGCTTACTTCCAT |
| | Probe | FAM- AGAAGCTGCTATTGCCA-TAM |
| pCeu vector | For | GTCCTTGTTATTTCCTGTCGTCGCTACCTTAGGACCGTTATAGT-TACGAGTAATCATGGTCATAGC |
| | Rev | CCTCAATTAGCAAGCGACAGTCGCTACCTTAGGACCGT-TATAGTTACGTCACTGGCCGTCGTTTTA |
| U70/73 | For | CTGTCGCTTGCTAATTGAGG |
| | Rev | CGACAGGAAATAACAAGGAC |
| Colony PCR DR1 | For | ACTCTGTGTTACACCACCT |
| | Rev | CTTGCTGTAAACATGACCA |
| ΔimpTMR | For | ATCCTAAATAACCCCCGGGGGGTAAAAAAGGGGGGGAGC-CAGGTCATCTGTTCTAGATCTAGGGATAACAGGGTAATCGATTT |
| | Rev | GTCAGGGCAGATATGGATAGGATCTAGAACAGATGACCTG-GCTCCCCCCCTTTTTTTACCGCCAGTGTTACAACCAATTAACC |
| ΔpTMR | For | AATCCTAGTCCGTAACCGTAACCCCAATCCTAGCCCTTAGC-CCCGCCCCCACTGGCAGCCTAGGGATAACAGGGTAATCGATTT |
| | Rev | TGAAGGCATTACAAGACATTGGCTGCCAGTGGGGGCGGG-GCTAAGGGCTAGGATTGGGGTGCCAGTGTTACAACCAATTA-ACC |
| impTMR (seq) | For | CTAGAGTTATTGCCGTGCGT |
| | Rev | ACTGTGATGCACATCATCAC |
| pTMR (seq) | For | TGCTAAATCCCCTGAAACAC |
| | Rev | GTGCTCTTTGTCTTTGCTCT |
| Lenti-vector | For | TCTAGACCCAGCTTTCTTGT |
| | Rev | GGATCCGAGTTTGTCGTCAT |
| CD4 | For | GGTTTTGGCAGTACATCAATGG |
| | Rev | TTTGTACAAGAAAGCTGGGTCTAGACTCGAGTCAATTAAG |
| U18 | For | GATGACGACAAACTCGGATCCATGGGAAAAGAGATGATGTT |
| | Rev | TACAAGAAAGCTGGGTCTAGATCAACCAACATTCTTGTCAACT |
| U53 | For | GATGACGACAAACTCGGATCCATGGAAACTGTGCTAGTGGC |
| | Rev | TACAAGAAAGCTGGGTCTAGATCAATTGTCCATTTTATTCAAAG |
| U73 | For | GATGACGACAAACTCGGATCCATGGAAACTCAGCTTCAAAAT |
| | Rev | TACAAGAAAGCTGGGTCTAGATTATGAACGCAATAGACACC |
| U90 | For | GATGACGACAAACTCGGATCCATGGAAAGAAGTGGAGCTAC |
| | Rev | TACAAGAAAGCTGGGTCTAGATTAATAGATTTGAGCATTTT |
| crRNA U7 | crRNA | ATCGGCTTCGCGAGATATTGTGG |
| crRNA U10 | crRNA | TAAAGTATCGTTCCTTTCGGTGG |
| crRNA DR6 #1 | crRNA | CAATCGAGCGACAGTCACCAAGG |
| crRNA DR6 #2 | crRNA | AACACACGTGACAGCTCGCACGG |

generated based on purified HHV-7 BAC DNA using the DIG-High Prime kit (Sigma-Aldrich) as described previously. The images were acquired using a Leica DMi 8 Widefield Microscope (Leica, 100× objective). At least 100 metaphases or 100 interphases (nuclei) were imaged per cell line and the integration frequencies quantified. The image J software (https://imagej.nih.gov/ij/) was used to analyze the images.

## Assessment of integration sites by Nanopore sequencing

High molecular weight (HMW) DNA was isolated from HHV-7 positive U2OS clones and used for nanopore sequencing [37]. Briefly, $10^6$ cells were washed twice with PBS. RNA and proteins were removed by RNase and proteinase K digestion. To completely remove the proteins, we extended the proteinase K digestion time to 16 h. Viral DNA in the supernatant was purified by phenol-chloroform and ethanol precipitation. The obtained HMW DNA was dissolved in nuclease-free TE buffer (IDT; pH 8.0) and used for nanopore sequencing [50,51]. To increase the number of reads on the virus genome, we developed a Cas9-based enrichment strategy for the HHV-7 genome [52]. Briefly, specific crRNAs were designed for the U7, U10, and DR6 region in the HHV-7 genome using the CHOPCHOP software (https://chopchop.cbu.uib.no/) (Fig 2A and Table 1). As the orientation of the crRNAs and the position of the PAM sequence define the directionality of sequencing, we designed crRNAs to target both the direct strand (probes crRNA DR6 #1 and #2) and complementary strand (probes crRNA U7and U10), thereby generating reads that extend toward the 3' and 5' ends of the viral genome, respectively.

Sequencing library were prepared according to the standard protocol (https://nanoporetech.com/en/document/cas9-targeted-sequencing#overview-of-the-protocol) with the following modifications. The genomic DNA was purified with 0.45x AMPure XP beads (Beckman Coulter) and eluted in 26 µL of nuclease-free water after incubation for 60 min at 37 °C. 5 µg HMW DNA was dephosphorylated at 37 °C for 20 min to prevent unspecific linker ligation and the enzyme was subsequently deactivated and 80°C for 4 min. The dephosphorylated DNA was then incubated with Cas9 and four specific guide RNAs to cleave the virus genome in the U7/U10 and DR6 region at 37 °C for 22 min, exposing phosphates for the nanopore linker ligation. The final library was loaded onto a MinION R9.4 flow cell, and data were collected as long as pores were available for sequencing. The raw sequencing data was assessed with Dorado (v. 0.7.2) using the basecalling model sup@latest option. Reads were mapped against the HHV-7 reference genome (NC_001716.2) and the human genome (T2T-CHM13v2.0) using minimap2 (v. 2.26) implemented in epi2me-labs/wf-alignment (v. 1.1.2). Read alignments from BAM files were visualized using IGV (v. 2.17.4). Only reads mapping to the HHV-7 DRs were used for analysis of viral integration sites.

## Generation of the first HHV-7 BAC by TAR cloning

HMW HHV-7 DNA (JI strain) was isolated from highly infected SupT1 cells as described above. Linear HHV-7 genomes were obtained by restriction digestion of Fast Digest FspAI (NEB). The YCp-BAC vector was derived from pCC1BAC-His3 [38] and published previously [26]. To visualize infected cells, an EGFP cassette driven by the HSV-1 TK promoter was cloned into the YCp-BAC vector. The homologue sequences (the U71/U72 region of HHV-7, see Fig 3C) were first cloned into the pCeu backbone [53] and subsequently transferred into the YCp-BAC vector. To obtain the linear YCp-BAC vector with the U71/U72 hooks, we performed a restriction digestion using I-CeuI (NEB). Alternatively, an inverse PCR was performed using the PrimeSTAR GXL DNA polymerase (Takara) with the primers in Table 1.

The HMW HHV-7 DNA and the linear YCp-BAC-U71/U72 vector was used to assemble the HHV-7-GFP BAC by single-step TAR cloning in the *Saccharomyces cerevisiae* strain VL6-48N (*MATα trp1-Δ1 ura3-Δ1 ade2–101 his3-Δ200 lys2 met14 cir°*) as described previously [54,55]. Briefly, yeast spheroplasts were obtained by digesting yeast cells with Zymolyase (Carl Roth). 5 µg pre-digested HMW HHV-7 genomic DNA and 250 ng linear YCp-BAC-U71/U72 vector were mixed and gently transformed into yeast spheroplasts. Colonies in SD/-His agar plates (Takara) were observed 72 h post-transformation. To screen the positive clones, DNA was extracted using lysis buffer containing fresh Zymolyase and

β-Mercaptoethanol (Carl Roth). The positive clones were screened by PCR-amplification using the primers in Table 1. PCR positive HHV-7-GFP BACs were transformed into the *E. coli* strain 10G (LGC Biosearch Technology).

## Generation of TMR mutant HHV-7

HHV-7-GFP BAC DNA was first electroporated into *E. coli* strain GS1783 (*DH10B lcI857 Δ(crobioA)<>araC-PBADIsceI*), which facilitates modification of BACs using the markerless "*en passant*" mutagenesis [27,28,56]. TMR mutants were generated based on the HHV-7-GFP BAC using the primers listed in Table 1. Recombinant BAC clones were confirmed by PCR, RFLP, Sanger sequencing (LGC Genomics). For whole BAC sequencing, DNA was prepared using the QIAGEN Plasmid Midi Kit (QIAGEN). Whole BAC nanopore sequencing of the BAC clones was performed by Plasmidsaurus (USA) and/or Eurofins (Germany).

To reconstitute the recombinant viruses, high-quality BAC DNA was extracted using the NuleoBond Xtra BAC extraction kit (Macherey-Nagel). Subsequently, $5 \times 10^6$ SupT1 cells were resuspended in 100μL Amaxa V buffer [57] and nucleofected with 3 μg BAC DNA and 1 μg pp71 DNA [58] using program U-014 of the Nucleofector 2b device (Lonza). To improve the reconstitution, we generated SupT1 cells overexpressing the HHV-7 proteins U18, U53, U73 or U90. These cell lines were generated by i) cloning these genes into the pLenti vector backbone [48], ii) production of these lentiviruses as described above and iii) transduction of the SupT1 cells. To improve virus replication, transfected SupT1 cells were stimulated with 9 μg/mL hydrocortisone, 3.75 μg/mL PHA (Roche), or 10 nM IOX2 (Sigma) and 1 ng/mL TPA (Roche) as shown in Fig 4A. After 24 h of TPA stimulation, cells were washed once with PBS, and pre-stimulated cells (hydrocortisone-treated for 24 h) were then added at a ratio of 2:1 (transfected cells to stimulated cells).

## Replication kinetics assay

To differentiate between the inoculum and target cells, we stained $2 \times 10^7$ SupT1 target cells with the CellVue Claret Far Red Fluorescent Cell Linker Mini Kit (Sigma). Target cells were infected with the virus stocks, isolated using a FACS Aria III cell sorter (BD) and infection was monitored for 6 days by qPCR. For HHV-7-GFP and the TMR mutant viruses, target cells were infected as described previously [11,59,60] and the percentage of GFP + was assessed using CytoFlex S FACS analyzer for 6 days. Data were analyzed using the CytExpert (Beckman Coulter) and FlowJo (FlowJo LLC) software.

## Statistical information

Statistical analyses were performed using GraphPad Prism 9.0.2 (GraphPad Software, LLC). Replication kinetics and integration efficiencies were analyzed using one-way ANOVA followed by Wilcoxon matched-pairs signed rank test and Dunnett's multiple comparisons test, respectively. Data were considered significant when p-value ≤ 0.05. All experiments have been repeated at least three independent times (N = 3). All figures were generated by the authors using Affinity Designer 2 and BioRender.

## Supporting information

**S1 Data. Raw sequencing reads obtained from HHV7-infected clonal cell lines using the Cas9-based enrichment strategy.**
(XLSX)

**S2 Data. Whole BAC nanopore sequencing of HHV-7 GFP and its TMR mutants.** All four BAC clones were shown as GenBank files.
(XLSX)

**S3 Data. Sanger sequencing of DR region of these efficiently replicating HHV-7 GFP and its TMR mutant viruses.**
DR regions were amplified using PrimeSTAR GXL DNA polymerase (Takara) with primers around impTMR region
(Primer-1 F: 5'-CTAGAGTTATTGCCGTGCGT-3' and Primer-1 R: 5'-ACTGTGATGCACATCATCAC-3'),
and primers around pTMR region (Primer-2 F: 5'-TGCTAAATCCCCTGAAACAC-3' and Primer-2 R:
5'-GTGCTCTTTGTCTTTGCTCT-3').
(XLSX)

**S1 Fig. HHV-7-GFP BAC virus reconstitution in SupT1 cells expressing HHV-7 U18, U53, U73, and U90.** Cells were
nucleofected with the HHV-7-GFP BAC and stimulated with drugs at 2 dpt. GFP expression level is shown for the indicated cell lines and time points. HHV-7-GFP could only be reconstituted in SupT1 expressing U18 or U73.
(TIF)

## Acknowledgments

We thank Ann Reum, Yu You, Giulia Aimola and Yvonne Weber for their technical assistance.

## Author contributions

**Conceptualization:** Yingnan Cheng, Benedikt B. Kaufer.

**Data curation:** Yingnan Cheng.

**Formal analysis:** Yingnan Cheng, Joilson Xavier.

**Funding acquisition:** Benedikt B. Kaufer.

**Investigation:** Yingnan Cheng, Joilson Xavier, Dusan Kunec, Jana Reich, Christiana Victoria Cismaru, Dilan Gün Serdar, Thomas Höfler.

**Methodology:** Yingnan Cheng, Joilson Xavier, Dusan Kunec, Jana Reich, Dilan Gün Serdar, Thomas Höfler.

**Project administration:** Benedikt B. Kaufer.

**Supervision:** Benedikt B. Kaufer.

**Validation:** Yingnan Cheng.

**Visualization:** Yingnan Cheng.

**Writing – original draft:** Yingnan Cheng, Benedikt B. Kaufer.

**Writing – review & editing:** Yingnan Cheng, Joilson Xavier, Dusan Kunec, Jana Reich, Christiana Victoria Cismaru, Dilan Gün Serdar, Thomas Höfler, Benedikt B. Kaufer.

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
