## [Decision Letter · Decision Letter 0]

15 Dec 2025

PPATHOGENS-D-25-02654

Human herpesvirus 7 integrates into host telomeres via its telomeric repeat arrays

PLOS Pathogens

Dear Dr. Kaufer,

Thank you for submitting your manuscript to PLOS Pathogens. After careful consideration, we feel that it has merit but does not fully meet PLOS Pathogens's publication criteria as it currently stands. Therefore, we invite you to submit a revised version of the manuscript that addresses the points raised during the review process.

We look forward to receiving your revised manuscript.

Kind regards,

Laurie T Krug, PhD

Academic Editor

PLOS Pathogens

Blossom Damania

Section Editor

PLOS Pathogens

Sumita Bhaduri-McIntosh

Editor-in-Chief

PLOS Pathogens

orcid.org/0000-0003-2946-9497

Michael Malim

Editor-in-Chief

PLOS Pathogens

orcid.org/0000-0002-7699-2064

**Additional Editor Comments:**

Each of the reviewers had major concerns about the submission. Overlapping concerns were the lack of novelty and impact on the field. Reviewers noted the need to provide additional data regarding the integration sites of mutant genomes and the structure of recombinant genomes after replication. The need for greater experimental details regarding the U20S cell infection system was also cited. Last, reviewers noted major revisions, both grammatical and stylistic, were needed to multiple sections of the text. Please note that since additional experiments and a major rewrite are expected, a full round of reviews would be required with no guarantee of acceptance.

**Journal Requirements:**

https://journals.plos.org/plospathogens/s/submission-guidelines#loc-parts-of-a-submission

4) We notice that your supplementary Figures are included in the manuscript file. Please remove them and upload them with the file type 'Supporting Information'. Please ensure that each Supporting Information file has a legend listed in the manuscript after the references list.

Potential Copyright Issues:

i) Figures 2A, 3A, 3B, and 6A. Please confirm whether you drew the images / clip-art within the figure panels by hand. If you did not draw the images, please provide (a) a link to the source of the images or icons and their license / terms of use; or (b) written permission from the copyright holder to publish the images or icons under our CC BY 4.0 license. Alternatively, you may replace the images with open source alternatives. See these open source resources you may use to replace images / clip-art:

6) In the online submission form, you indicated that your data will be submitted to a repository upon acceptance. We strongly recommend all authors deposit their data before acceptance, as the process can be lengthy and hold up publication timelines. Please note that, though access restrictions are acceptable now, your entire minimal dataset will need to be made freely accessible if your manuscript is accepted for publication. This policy applies to all data except where public deposition would breach compliance with the protocol approved by your research ethics board. If you are unable to adhere to our open data policy, please kindly revise your statement to explain your reasoning and we will seek the editor's input on an exemption.

7) In the online submission form, you indicated that The HHV-7-GFP BAC and TMR mutant viruses are available upon request and were sequenced by nanopore sequencing.. All PLOS journals now require all data underlying the findings described in their manuscript to be freely available to other researchers, either

1. In a public repository

2. Within the manuscript itself

3. Uploaded as supplementary information.

8) Please amend your detailed Financial Disclosure statement. This is published with the article. It must therefore be completed in full sentences and contain the exact wording you wish to be published.

**Reviewers' Comments:**

Reviewer's Responses to Questions

**Part I - Summary**

Reviewer #1: Cheng et al. demonstrated integration of human herpesvirus 7 (HHV-7) genome into chromosome in CD4 expressing U2OS cells by using fluorescence in situ hybridization (FISH) analysis and Nanopore sequencing. The integration pattern is consistent with those of the previously known viruses, such as HHV-6A/B and Marek’s disease virus, which also harbor the terminal telomeric repeat. In addition, authors have cloned the HHV-7 genome as a BAC-maintained DNA in yeast and demonstrated modification of genome in yeast, virus reconstitution from the recombinant DNA through refined methods, and the integration of recombinant genome into the chromosome in U2OS cells. Using the recombinant HHV-7 system, authors revealed that the absence of either the perfect telomeric repeat (pTMR) or imperfect telomeric repeat (impTMR) alone has little effect on the integration, but lack of both significantly diminish the integration efficiency.

This study clarified the possibility of the genome integration of HHV-7, a pathogen latent in our body with many unknown aspects. Experimental techniques, especially the establishment of HHV-7 recombinant system, are excellent. Although most of the findings presented in this manuscript are highly predictable considering the similarity of the genome structures with known viruses, the collected dataset and the novel recombinant system are regarded as highly valuable. In terms of novelty, the authors pointed out one unique feature of the genome integration of HHV-7; “loss of the impTMR can be complemented by the pTMR, and vice versa” (line 299), in contrast to those of HHV-6A/B and MDV. By taking one more step in the analysis, their manuscript will be more valuable for the publication.

Reviewer #2: Betaherpesviruses are endemic in the general population with three distinct members involved in human infection. Of these, HHV-7 remains the most understudied of its siblings, HCMV and HHV6a & b. As such, a significant amount of virologic investigations have yet to be performed on this pathogen. In this manuscript, the authors seek to determine if HHV-7 can establish an infection resulting in a viral genome being integrated at the telomeric ends of human chromosomes similar to that found with its roseolavirus counterparts. To this end, the investigators have generated a stable cell line expressing the HHV-7 receptor (CD4) and can observe long term genomic maintenance of HHV-1 in about 10% of the cell population after a month in which the genome copy number to cell number is ~1. FISH analysis confirmed genomic integration of HHV-7 DNA at the ends of chromosomes at regions that contain telomeres, a finding that was anticipated as the viral genome contains perfect and imperfect repeats of telomeric sequences. Isolation of viral integrated cell lines were further validated by genomic sequencing which revealed that the HHV-7 genome was integrated at telomeric ends in which recombination occurred that deleted one copy each of the Pac1 and Pac2 sequences which are needed for circularization of the genome upon infection. In order to determine the requirement of the perfect and imperfect viral telomeric regions for host cell integration the authors generated a BAC recombinant of HHV-7 by exploiting Yeast recombination of linearized virus followed by bacterial recombination protocols to manipulate the viral genome. Mutants lacking either the perfect telomers, the imperfect telomers or both were generated. While the viral telomeric repeats were found to be expendable for lytic replication of the virus, the double deleted telomeric repeat virus failed to exhibit host cell integration whereas the WT, and each of the single mutants were still capable of integration. The authors conclude that either telomeric region can suffice for viral integration yet a deletion of both renders the virus incapable of integration.

Overall this is a well written study and the conclusions are supported by the data presented. That being said, the overall findings are not particularly surprising as one would imagine HHV-7 would behave very similar to its closely related counterparts. While the generation of the BAC containing HHV-7 infectious construct is of value to the community, the study is mostly confirmatory to what one anticipates from roseoloviruses. Some experimental suggestions are provided below.

Reviewer #3: Human herpesvirus 7 (HHV-7) is a member of the Roseolovirus genus in the betaherpesvirus subfamily. Most adults have been infected since early childhood, and harbor the usually benign virus for the remainder of their lives. HHV-7 genome architecture is similar to that of other roseoloviruses, including human herpesviruses 6A and 6B (HHV-6A and (HHV-6B). An interesting shared feature is the presence of DNA sequences at their genomic termini that are similar to sequences present at the termini of chromosomes of many eukaryotes, (GGGTTA)n. These sequences enable the virus genomes to be incorporated into the chromosomes of cells they infect, including germline cells. About 1% of human are born with inherited chromosomally integrated (ici) genomes of HHV-6A or HHV-6B (iciHHV-6A and iciHHV-6B). Mechanisms and outcomes related to these interesting and intriguing activities are being investigation in several laboratories.

In this paper, Cheng and colleagues conducted a series of experiments that clearly demonstrate that HHV-7 integrates into human chromosomes via its telomeric repeat arrays.

This is important work. It includes a technically sophisticated and well-executed experimental design. I don’t have a lot of comments on the design, execution, and illustration of the experiments. Unfortunately, the text needs a lot of work. I am providing a detailed list of suggested edits for pages 1-4. It is likely that when these items are repaired, other issues will become visible in the same area, so the authors need to look beyond the current comments prior to resubmitting. While I did not take the time to document everything, the high density of text issues extends through the rest of the text.

**Part II – Major Issues: Key Experiments Required for Acceptance**

Reviewer #1: 1. According to the Nanopore sequencing data in Fig 2, the typical integration pattern of HHV-7 genome uses both impTMR and pTMR keeping the both of DRL and DRR, as illustrated in Fig 2C. However, the recombinant HHV-7 genome lacking either impTMR or pTMR alone still can be integrated, as authors demonstrated in Fig 6. In either case, one of the DRL and DRR may be excluded after integration. As this may be an unexpected and unique feature of HHV-7, authors should provide information regarding the integration patterns for these mutant genomes to further confirm that genome integrations occur in similar or unexpected manners in cases of these mutants.

Reviewer #2: 1) some experimental specifics are in need of more details. For example, what was the MOI used to infect the U2OS-CD4 cells? What % were positive? Were the cells productively infected? Do they produce cell free virus? Do herpesvirus DNA replication inhibitors limit DNA amplification? Are lytic transcripts (rt-qPCR) or proteins (WB) observable? Are the low genome numbers seen at 28 days a result of integration or few lytically infected cells? Can the virus be reactivated from the clonal cell lines?

2) it was not clear if the integration sites are found in preferential host chromosomes or just randomly integrated in the telomers of any chromosome.

3) the discussion is just a rehash of the results section and not a comprehensive evaluation of the conclusions that one can derive from the findings.

Reviewer #3: As noted above, the paper needs significant re-writing to be able to judge its technical merits. To the extent that I could understand the writing, with good re-writing, the technical aspect of the paper will likely stand as Very Good or Excellent.

**Part III – Minor Issues: Editorial and Data Presentation Modifications**

Reviewer #1: 1. Fig. 1D and Lines 85-86, the copy numbers of 74-2 and 74-18 seem to be less than one. Is there any possibility, e.g. one half? Alternatively, does the data indicate the unstable integration of HHV-7 genome?

2. Fig. 2B, the loss of pac2 sequence is difficult to understand from this figure. Readers don’t know whether the label “pac2 sequence” is given for the sequence above it or the sequences below it. It could be confirmed after I zoomed in and read the reference sequence. Additional labels are helpful to indicate pTMR before the pac2 sequence, and to indicate pTMR and Telomere for actual sequences e.g. Sample 22-14.

3. I wonder whether the recombinant HHV-7 genomes can restore the linear genome structures as written in Fig 5A. The BAC DNA becomes circular form by connecting the ends of genome DNA i.e. pac1 and pac2, and it must be properly processed during reconstitution and lytic replication processes. Since the modification sites of ΔpTMR, ΔimpTMR, and ΔTMR are near the end of genomes, it may affect on the concatemer processing. Is there any data or information by which you can confirm the genome structure of each recombinant genome after replication?

Reviewer #2: (No Response)

Reviewer #3: The number and nature of text modifications needed takes this well beyond “minor”.

PLOS authors have the option to publish the peer review history of their article (what does this mean?). If published, this will include your full peer review and any attached files.

Reviewer #1: No

Reviewer #2: No

Reviewer #3: No

**Figure resubmission:**
---

## [Decision Letter · Decision Letter 1]

9 Mar 2026

PPATHOGENS-D-25-02654R1

Human herpesvirus 7 integrates into host telomeres via its telomeric repeat arrays

PLOS Pathogens

Dear Dr. Kaufer,

Thank you for submitting your manuscript to PLOS Pathogens. After careful consideration, we feel that it has merit but does not fully meet PLOS Pathogens's publication criteria as it currently stands. Therefore, we invite you to submit a revised version of the manuscript that addresses the points raised during the review process.

We look forward to receiving your revised manuscript.

Kind regards,

Laurie T Krug, PhD

Academic Editor

PLOS Pathogens

Blossom Damania

Section Editor

PLOS Pathogens

Sumita Bhaduri-McIntosh

Editor-in-Chief

PLOS Pathogens

orcid.org/0000-0003-2946-9497

Michael Malim

Editor-in-Chief

PLOS Pathogens

orcid.org/0000-0002-7699-2064

**Additional Editor Comments:**

Thank you for the thorough rebuttal and revisions that largely address reviewer concerns and improve the manuscript. There were a few issues that remain regarding a figure legend, integration frequencies by FISH, and incorporating other findings or differences with HHV-6 integration in your discussion. Please address in your resubmission and those text revisions will be quickly reviewed at the editorial level.

**Journal Requirements:**

1) In the online submission form, you indicated that The HHV-7-GFP BAC and TMR mutant viruses are available upon request (via email to virologie@vetmed.fu-berlin.de) and were sequenced by nanopore sequencing. . All PLOS journals now require all data underlying the findings described in their manuscript to be freely available to other researchers, either

1. In a public repository

2. Within the manuscript itself

3. Uploaded as supplementary information.

2) Please ensure that the funders and grant numbers match between the Financial Disclosure field and the Funding Information tab in your submission form. Note that the funders must be provided in the same order in both places as well.

3) Please make sure to include the correct citation for Biorender in the legends of Figures 2A, 3A,B,C, 4A,6A.

**Reviewers' Comments:**

Reviewer's Responses to Questions

**Part I - Summary**

Reviewer #1: Cheng and colleagues provide convincing evidence that human herpesvirus 7 (HHV-7) can integrate into host telomeres in persistently infected cells, combining fluorescence in situ hybridization with long-read nanopore sequencing. A major strength of this work is the establishment of the first HHV-7 reverse genetics platform, which will be broadly useful to the field. Using this system, the authors further dissect the contribution of the perfect and imperfect telomeric repeat arrays, showing that these elements are dispensable for lytic replication but are critical for efficient integration and for long-term genome maintenance during persistent infection. Although several aspects of the biology are conceptually consistent with what is known for other roseoloviruses, the datasets are solid and the technical advances and resources provided here represent a clear contribution to the community. Importantly, the authors have also responded appropriately to the comments raised by the three reviewers and improved the manuscript accordingly. Overall, the conclusions are supported by the data, and I recommend acceptance after minor revision to further improve clarity of presentation and the framing of novelty (particularly in comparison with HHV-6A, HHV-6B, and other telomere-associated herpesviruses), and to address a small number of remaining points that can be handled with limited additional work.

Reviewer #2: The authors have sufficiently addressed any concerns raised in initial review. The modified manuscript is sufficiently improved and no further issues remain.

**Part II – Major Issues: Key Experiments Required for Acceptance**

Reviewer #1: 1. The study shows HHV-7 telomeric integration in a cell-line model, but the manuscript does not adequately address how often HHV-7 integration is detected in human genomes. In contrast to HHV-6A/6B (for which inherited integration is reported at a measurable global frequency), HHV-7 integration in humans is scarcely documented (the cited Prusty et al., JGV 2017 reports only two cases). If population-level estimates or screening data exist, please cite and summarize them in the Introduction/Discussion. If not, explicitly state this limitation and discuss plausible reasons why HHV-7 integration is not maintained or detected in humans at HHV-6A/6B-like frequencies.

2. Figure 6B provides key evidence for integration, but the frequency of the shown FISH pattern is not quantified for each mutant. Please report the percentage of cells displaying the integration-associated pattern for each genotype. In addition, compare the HHV-7 integration efficiency with that reported for HHV-6A/6B in comparable U2OS-based systems (e.g., ~5–20% in Gravel et al., JVI 2017) and discuss possible reasons for any differences.

Reviewer #2: None.

**Part III – Minor Issues: Editorial and Data Presentation Modifications**

Reviewer #1: 1. (Fig. 6C): The presentation/legend needs clarification. The plot appears boxplot-like, yet the legend states that “mean HHV-7 genome copies … of 3 independent experiments” are shown. Please revise the legend (and figure if needed) to explicitly define what the center line and box/whiskers represent (mean vs median; SD/SEM vs range/IQR), and indicate n=3. Ideally, show the individual data points for the three experiments.

Reviewer #2: None.

PLOS authors have the option to publish the peer review history of their article (what does this mean?). If published, this will include your full peer review and any attached files.

**Do you want your identity to be public for this peer review?** For information about this choice, including consent withdrawal, please see our Privacy Policy.

Reviewer #1: No

Reviewer #2: No

**Figure resubmission:**
---

## [Editor Report · Decision Letter 2]

20 Apr 2026

Dear Dr. Kaufer,

We are pleased to inform you that your manuscript 'Human herpesvirus 7 integrates into host telomeres via its telomeric repeat arrays' has been provisionally accepted for publication in PLOS Pathogens.

Best regards,

Laurie T Krug, PhD

Academic Editor

PLOS Pathogens

Blossom Damania

Section Editor

PLOS Pathogens

Sumita Bhaduri-McIntosh

Editor-in-Chief

PLOS Pathogens

orcid.org/0000-0003-2946-9497

Michael Malim

Editor-in-Chief

PLOS Pathogens

orcid.org/0000-0002-7699-2064
---

## [Editor Report · Acceptance letter]

Dear Dr. Kaufer,

We are delighted to inform you that your manuscript, "Human herpesvirus 7 integrates into host telomeres via its telomeric repeat arrays," has been formally accepted for publication in PLOS Pathogens.

Best regards,

Sumita Bhaduri-McIntosh

Editor-in-Chief

PLOS Pathogens

orcid.org/0000-0003-2946-9497

Michael Malim

Editor-in-Chief

PLOS Pathogens

orcid.org/0000-0002-7699-2064